# Review of Recent Progress on Advanced Photocathodes for Superconducting RF Guns

**DOI:** 10.3390/mi13081241

**Published:** 2022-08-02

**Authors:** Rong Xiang, Jana Schaber

**Affiliations:** 1SRF-Gun Group, ELBE Department, Institute of Radiation Physics, Helmholtz-Zentrum Dresden-Rossendorf, 01328 Dresden, Germany; j.schaber@hzdr.de; 2Institute of Physical Chemistry, Technische Universität Dresden, 01062 Dresden, Germany

**Keywords:** photocathode, quantum efficiency, superconducting RF photoinjector, SRF gun

## Abstract

As is well known, the quality of the photocathodes is essential for the stability and reliability of photoinjector operations. Especially for superconducting radio frequency photoinjectors (SRF guns), the photocathode represents one of the most critical parts. Benefiting from the fast development of photocathode technology in recent years, several SRF guns have been successfully operated or tested for beam generation at the kHz–MHz repetition rate. In this paper, we will review the achievements as well as the open questions in the applications of photocathodes for SRF gun operation. Furthermore, we will discuss the possible improvement of photocathodes for future CW electron sources.

## 1. Introduction

The superconducting radio frequency photoinjector (SC RF gun or SRF gun) combines the principle of a photoelectron gun with the application of a superconducting (SC) accelerating cavity. The photocathode is located very close to the SC cavity or directly in contact with it. Since the first concept was proposed by H. Chaloupka in the 1988 EPAC [1], the SRF gun has been proved to be a successful technology producing high brightness and a high-current beam, which is required by continuous-wave (CW) free electron lasers (FELs) and energy recovery linac (ERL) facilities [2,3,4,5]. Recently, more and more attention has been paid to this type of modern electron source. Compared to the “traditional” direct current (DC) guns [6,7,8] and warm radio frequency (RF) guns [9,10], this type of injector is even more challenging, attractive, and promising.

Photocathode quality is a critical key issue for the stable operation of SRF guns. In fact, the photocathodes and the superconducting (SC) cavities are both extremely sensitive. Therefore, the first question for an SRF gun designer should be how to combine the two parts, which defines the main working concept of a gun. There are several reasonable solutions to integrate the cathodes with the SC cavities: (1) to insert a superconducting material, such as robust niobium (Nb) or lead (Pb), into the cavity back wall. (2) To integrate a normal conducting (NC) cathode into SC RF guns with an RF choke or behind a DC gap.

Thanks to the fruitful photocathode studies of recent decades, various options exist for cathode materials for SRF guns. However, besides the general requirements of photocathodes, such as high enough quantum efficiency (QE), long lifetime, low thermal emittance, fast response time, low field emission, and particle-free surfaces, there are also special requirements for the environment of SRF guns. (1) Sufficient thermal properties at cryogenic temperatures, such as thermal conductivity, thermal expansion, and temperature change through laser heating and RF power deposition. (2) Reducing possible multipacting (MP) induced by the cathode. (3) Superconducting properties for SC cathodes, such as transition temperature and critical magnetic field.

Based on this background, it is worth making an overview of the recent progress on photocathode studies for SRF guns, especially with a focus on the operational aspects, such as their advantages and limitations for this specific application. We start with the cathode requirements and the integration methods into the SRF gun environment and then introduce the SC photocathodes and the NC photocathodes applying for SRF guns. At the end, we summarize the present status of state-of-art photocathodes in application to SRF guns.

## 2. Requirements for Photocathodes Used in SRF Guns

In general, photoinjectors request that the cathode materials provide enough QE, practical lifetime, low thermal emittance, and fast response time. For practical operation in the electric field and for machine safety reasons, the cathodes need to be particle free, have smooth surfaces, good adhesive features, and low dark current.

Moreover, there are some special requirements for the photocathodes used in SRF guns. Because the cathode will be inserted into or close to the superconducting environment, it needs to be cooled in order to reduce the heat load to the SC cavity. So, special attention must be paid to the material properties at different temperatures, e.g., the thermal conductivity and thermal expansion. Especially when a photocathode is in direct contact with a gun cavity, its transition temperature and critical magnetic field must match the SC working conditions. When the temperature of semiconductor photocathodes is cooled from warm down to cryogenic temperatures, there will be an influence on the crystal’s lattice and the carriers’ density. Therefore, it may change the energy band gap and electron affinity, which directly influence the QE and intrinsic emittance at a certain laser wavelength [11].

The heat load from the cathode to the cavity should be taken into account. A part of the cathode drive laser and the RF power deposited on the cathode surface can cause additional heat. If no sufficient cooling is provided for the cathode, this heat might warm up the cathode and even affect the SC cavity. Therefore, the material used for the cathode structure must ensure good thermal conductivity and good thermal contact with the cooling circular.

MP is an unwanted phenomenon limiting the SC cavities’ performance in both elliptical cavities and quarter-wave resonator (QWR) cavities. MP appears when the field-emitted electrons start to resonate in the RF field and the electron numbers are amplified. Thus, the cathodes used in SRF gun cavities need to have a low secondary electron yield (SEY) in order to effectively reduce the MP possibility.

## 3. Photocathode Assembly

As already mentioned, the integration of a photocathode into an SC cavity represents the key technology for an SRF gun. Up to now, several cathode assembly methods have been successfully applied in SRF gun designs.

The superconducting photocathode can be directly sealed onto the cavity or it is a part of the cavity back wall. In this case, no special assembly method is needed. For example, the DESY SRF gun [12], JLab SRF gun [13], and Euclid UEM SRF gun [14]. The disadvantage is that the cathode cannot be exchanged or is not exchanged in situ.

However, if an NC photocathode is chosen as a photoemitter, there must be thermal isolation between the NC cathode and SC cavity, for example, a vacuum gap. Several RF choke structures have been designed for the cathode holder (stalk) in order to minimize the RF loss to this vacuum gap. The structure of the RF choke cavity is suitable for the 1.3 GHz elliptical cavity guns [5,15,16]. BNL [17] and WIFEL SRF gun [18] cathode stalk structure is used for the low-frequency QWR cavity gun. If the cathode itself is also electrically isolated from the cavity, a DC bias is able to be loaded on the cathode in order to suppress the possible MP resonance. Because of this vacuum gap, it is also possible to operate the cathode at different temperatures, e.g., cooled with liquid helium (KEK SRF gun), liquid nitrogen (HZDR SRF gun), cold gases (HZB SRF gun, SLAC SRF gun), or even with water (BNL SRF gun).

There is another solution to integrate the NC cathode by using a DC gun outside of the SC cavity, for example in the PKU DC-SRF gun. The vacuum gap between the cathode and cavity is extended until the cathode is located outside of the SC cavity, where the biased cathode and the cavity build a new Pierce DC gun. The Pierce-structure DC part can be kept at room temperature or cooled down to reduce the heat load. Compared to the other RF choke design, the main advantages to moving the cathode out from the cavity into an extra DC structure are little risk of contamination to the SC cavity and no RF heating to the cathode. The disadvantages are the limited DC field on the cathode surface and the drift space between the Pierce DC part and the cavity, which makes this design not suitable for high bunch charge applications.

## 4. Superconducting Photocathodes

As mentioned in Section 2, it is possible to choose a superconducting photocathode for an SRF gun to provide enough photocurrent when illuminated with a drive laser but bring no quenching problem to the cavity. The key challenges are to achieve high enough QE by removing the surface contaminations on the cathode and to operate the cathode without changing the SC cavity performance. We introduce niobium, lead, and the plasmonic enhanced cathode based on them in this section. The idea of a thin film multi-alkali photocathode on the SC substratum will be introduced in Section 5.4.

### 4.1. Nb Photocathode

The QE of a bare Nb surface can exceed 10^−5^ at 266 nm [19]. The most direct solution is to illuminate the back wall of an Nb cavity to generate electron beams, especially for low-current applications, such as CW ultrafast electron diffraction/microscopy (UED/UEM) [14], where only <1 pC bunch charge is required. Compared to the exchangeable plug-in photocathode, this method can avoid any possible mechanical difficulties and particle issues.

The design of the Euclid SRF gun is a good example (Figure 1). This is a CW, 1.5-cell SRF gun operating at 1.3 GHz for MeV UED/UEM applications. The cavity itself is coated with Nb_3_Sn, but the back wall center is bare Nb with a high residual resistivity ratio (RRR) used as a photocathode. An ultra-violet (UV) laser with milliWatts power illuminates the back wall of the SRF cavity to provide the required bunch charge ≤ 500 fC. The back wall photocathode must be robust because no exchange cathode is available. Of course, the metal photocathode is more robust against vacuum spikes and ion back-bombardment than the semiconductor cathodes.

Because the required laser power is very low, the heat effect on the local area is limited. The cooling efficiency through the back wall should be good enough to maintain the superconducting status at the cathode area.

### 4.2. Pb Layer on Nb

A project of all superconducting RF guns at DESY belongs to the CW-XFEL project [3]. It has a 1.6-cell, 1.3 GHz superconducting gun cavity with a lead cathode located in its back wall. Because a high bunch charge up to 1 nC is required for the XFEL facility, with the existing laser system, the QE of niobium is too low [19]. However, the QE of another superconductor, lead, is sufficiently high [20]. A test with the Pb-coated cathode plug has been conducted with an HZB SC RF gun [21].

In the work of [21,22], the lead film was deposited in the cathodic arc and then was treated with an ex situ smoothing process, re-melting and recrystallizing with pulsed plasma ion beam bombardment, followed by in situ laser cleaning treatment. This smoothing process could improve the surface conditions. The lead layer showed a very good QE of 2.7 × 10^−3^ at 213 nm after the laser cleaning.

The DESY SRF gun cavity and its cathode are still under development. The first cavities with a lead-coated back wall suffered from problems with cathode surface quality [12]. The cavity performance with a coated Nb plug was degraded compared to the test with an uncoated Nb plug. A recent attempt was made to use the Nb cathode plug screwed into the hole at the cavity back wall (Figure 2). The plan was to clean the coated cathode surface with a laser after cathode insertion in the clean room. However, there is a high risk that the lead particles falling down from the cathode might heat and quench the cavity. Another challenge is the exchange progress of the cathode. The cathode exchange must be performed when the cavity is warmed up and sent back to the clean room. Thus, from the practical operation side, it is only reasonable when the cathode can provide a year-long lifetime.

However, more experimental results are needed to verify the feasibility and reliability of this design. The superconducting status at the cathode area can be maintained only when the heat power from the drive laser is limited and can be efficiently cooled by the back wall.

### 4.3. Nb Cathode with Plasmonic Enhancement

A metallic photocathode with a nano-patterned surface will obtain plasmonic effects to increase the absorption of light and enhance the local electromagnetic field intensity [23,24]. This effect can be also used for the superconducting photocathode material, for example, niobium. The team from Jlab and RadiaBeam has proposed a promising new design with a 1.6-cell 1.3 GHz cavity and a special cathode plug cooled with liquid helium (Figure 3).

There are different approaches to obtaining the plasmonic enhanced surface. One method is to fabricate a rectangular array of subwavelength nanostructures with a focused ion beam (FIB) milling onto the metallic surface. This designed nano-pattern can excite properly tuned surface-plasmon response. In the experiment using nano-patterned niobium at room temperature (typical work function ~4.2 eV ~296 nm) [13], the measurement of charge density versus 1030 nm laser intensity suggested that four-photon emission appeared, corresponding to an effective QE of 1.4 × 10^−5^.

Another method described in [25] is to deposit a low work function material, such as magnesium (Mg), onto the Nb surface and ultra-thin islands of indium (In), which produces the deep UV plasmon resonance and improves the QE by >400 times.

This technique is a promising step to realizing high-QE superconducting photocathodes. At the moment, no reports about the operation performance of SRF guns have been published. There is a potential risk of quenching due to high laser power on the cathode. For the plasmonic cathode, the number of photoelectrons increases nonlinearly with the photon numbers. The higher the laser pulse energy, the higher the achieved QE. A balance must be reached between enough laser energy for photoemission and the heat load to the cathode. Thus, the maximum repetition rate of the electron source might be limited by the cathode quality. Furthermore, the cooling of the cathode area is an essential key for successful operation in the future. The special cooled plug design like at JLab (in Figure 3) is needed to enable cathode treatment outside of the cavity and to enhance the thermal conductivity with proper plug materials.

## 5. Normal Conducting Photocathodes

The application of normal conducting photocathodes in normal conducting RF guns and in DC guns is a mature technology. From Section 3, an SRF gun can also integrate an NC cathode without causing quality degradation for the SC cavity. At the moment, the NC photocathodes that are already operated or tested in SRF guns are metallic photocathodes and some semiconductor cathodes, such as Cs_2_Te, K_2_CsSb, and GaAs. In addition, there are new candidates for SRF gun applications, such as GaN.

### 5.1. Metallic Photocathodes: Cu, Mg

As is well known, metallic cathodes are robust and easy to treat and transport. Most of them have good RF properties and thermal performance. Moreover, they show a low risk for cavity contamination. Thus they are always good choices for commissioning a new gun in which cathode robustness is highly required. For example, a copper cathode was inserted into the cavity during the commissioning of the HZDR SRF Gun-II [26] and HZB SRF gun [16].

Metals with lower function can relieve the drive laser. Theoretically, stable metals with a work function of less than 5 eV (laser wavelength 248 nm) can be considered as photoemitters. Table 1 lists the metals recently reported for SRF guns. In practice, copper is a favorable cathode material with a moderate work function of 4.6 eV. Cu photocathodes have very good RF performance, low dark current, and low second electron yield [27], which can greatly help to relax the difficult RF commissioning process. On the other hand, its typical QE is in the range of 10^−5^ at 258 nm, which provides enough electrons for the first beam diagnostic purpose.

Another suitable candidate as a photocathode for SRF guns is magnesium. It has a low work function of 3.6 eV and a QE up to 0.5% with 258 nm UV light, which is the highest QE ever reported for metal photocathodes. In HZDR SRF Gun-II, Mg has been successfully used for high current user operation since 2016 [26]. The experiments proved that Mg is able to provide high QE with a long lifetime, low dark current, and only minor MP problems. A cleaned Mg cathode can be stored in a vacuum at 10^−10^ mbar without detectable QE dropping, and it can be operated in an SRF gun with stable performance for a year long. Because there is no active alkali metal on the surface like the cesiated semiconductor cathodes, the risk of MP original from the cathode material is very low.

In the HZDR SRF gun, the Mg cathode consists of a bulk plug-in cylinder with a 10 mm diameter, which is operated at LN_2_ temperature. The plug of pure 99.9% poly-crystalline Mg has a mirror-like polish. After the chemical de-oxidation process, the QE of the Mg cathodes is only about 2 × 10^−5^. Then, the picosecond UV drive laser is used for the in-vacuum cleaning of the Mg surface with a 100 kHz repetition rate and 100 mW power in the dedicated cathode transport chamber. The laser cleaning leads to a QE improvement of two orders of magnitude, achieving values between 0.2 and 0.5 %. The cleaning process is very well reproducible. The QE stays stable for a very long time in the transport chamber with a vacuum 10^−10^ mbar as well as during SRF gun operation. The procedure is to exchange the transport chamber at the gun once a year during the shut-down time. The cathode always has a shiny, silver color, but the surface morphology changes remarkably, as shown in Figure 4. The virgin part is the polished mirror-like surface while the cleaned part shows a periodic structure, which is created by the scanning pattern of the laser beam. More information can be found in reference [29].

The Mg photocathode is expected to have low thermal emittance (0.5 µm/mm) because the surface photoemission plays an important role for Mg compared to those high QE photocathodes with volume photoemission [30]. On the other hand, the surface produced by laser cleaning can increase the thermal emittance. More study is required to clarify this effect.

### 5.2. Cs_2_Te for HZDR SRF Guns

Semiconductor photocathode Cs_2_Te is already an established material used in normal conducting RF guns, such as DESY [31], CERN [32], ANL [33], KEK [34], and so on, mainly benefiting from its high robustness. The response time of Cs_2_Te has been measured in RF guns shorter than 400 fs [35,36], which is fast enough for L-band RF guns.

Cs_2_Te is also suitable for SRF gun applications. For example, Cs_2_Te is chosen as the photocathode for the high bunch charge operation mode of HZDR. Before 2014, Cs_2_Te on Mo was successfully operated in the SRF gun-I [37], and this gun was applied as the e-source for IR-FEL application [38]. However, during the CW beam operation in SRF Gun-II in 2017, the bad thermal contact between the Mo plug and Cu holder in the gun led to a serious overheating problem. Consequently, the plug material was changed to copper to ensure better thermal conductivity for LN_2_ cooling, although copper is not a friendly substratum for Cs_2_Te coating. Since 2019, Cs_2_Te on copper has been operated as the standard photoemitter [39]. Each cathode provided about three months of beam with a 7–17 coulomb charge depending on the current request of the beam users.

The Cu plug is mirror-polished or finished with diamond turning to achieve a roughness at a level of 10 nm. Chemical cleaning, dry ice cleaning, and 350 °C baking in a vacuum are applied to keep the surface free of any pollution. During the preparation process, the plug is kept at 120 °C with halogen light, and illuminated with a 260 nm LED light. The tellurium and cesium are deposited subsequently through a ϕ 4 mm mask onto the plug surface. Films with different Te thicknesses from 3 nm to 10 nm are prepared for application. The prepared cathodes are stored in the transport chamber with a vacuum of around 5 × 10^−11^ mbar. Up to six photocathodes can be transported together from the photocathode clean room to the SRF gun.

Nearly every 3 months, the cathode is changed due to the low QE or poor QE distribution. The cathode QE before and after installing it into the gun is controlled routinely with a laser of 0.1 mW low power. When a new cathode is inserted into the gun, the MP around the cathode structure is one critical issue, leading to vacuum instability and thus to the first QE degradation. Furthermore, when the CW RF is loaded, the QE drops down to typically about 1% (Figure 5). However, sometimes the QE distribution is still possible to change dramatically during CW beam extraction, which strongly influences the beam emittance due to the inhomogeneous thermal emittance and space charge effect. The thermal emittance of Cs_2_Te in the HZDR SRF gun (0.6–1.2 µm/mm) is found strongly coupled to the QE distribution. The area with higher QE shows higher thermal emittance.

The QE degradation mechanism for Cs_2_Te in an SRF gun might work in the following way. The first reason is that the dark current or the beam halo hit the cold cavity wall and release the absorbed residual gases, which results in the vacuum increasing and cathode pollution. Secondly, the released gas molecules may be ionized by accelerated photoelectrons/dark current. Those ions might be accelerated by the RF field as well as the cathode DC field and then fly back to the cathode surface. However, theoretical work is still needed to understand this process.

The way to further push the cathode quality and the operation lifetime, on one hand, is to improve the cathode robustness, and on the other hand, is to modify the RF starting-up process to avoid MP and vacuum issues.

### 5.3. Reflection-Mode K_2_CsSb

The band gap plus electron affinity of K_2_CsSb is low, only 1.9 eV [40], which can be driven with a green laser, so it can relax the requirement of laser technology and save the laser cost. Meanwhile, it is reported as a material with low intrinsic emittance [41]. The advantages make K_2_CsSb a favorite cathode for high brightness photoinjectors. However, this photocathode is more sensitive to residual gases compared to Cs_2_Te. The storage and operation request an extremely good vacuum at a level of 10^−11^ mbar. The cryo-temperature in an SRF gun is able to fulfill this vacuum requirement, so several SRF gun projects are using or planning to use this photocathode. BNL, PKU, and HZB are using K_2_CsSb cathode in reflection mode, where the laser illuminates the cathode from the beam line side. KEK is developing K_2_CsSb on a transparent substrate so that the laser can illuminate the cathode from the backside.

#### 5.3.1. K_2_CsSb in BNL 113MHz SRF Gun

A reflection-mode K_2_CsSb photocathode has been successfully operated in the BNL SRF gun with high QE and a months-long lifetime, which benefits from the room temperature photocathode, avoiding the risk of decreasing QE due to low temperature. This concept obviously creates an extra heat load from the warm cathode to the SC cavity. A BNL SRF gun is able to deliver CW beams with a nano-coulomb bunch charge, providing a good example of high-charge, high-brightness CW electron beams [17,42]. The core design of this QWR SRF gun is shown in Figure 6. The cathode is installed inside of the water-cooled cathode stalk with a long manipulator. A gold-plated RF spring between the cathode and cathode stalk and a half-wave RF choke with an impedance transformer are used to prevent RF leaks into the stalk structure.

E. Wang described the detailed process of the K_2_CsSb film growth [43]. The K_2_CsSb is deposited with a dedicated rapid sequential cathode growth procedure on a polished Mo substrate puck. It is noteworthy that from their experience, the key to producing a stable, high-QE K_2_CsSb photocathode is the accurate control of the K-layer thickness. Furthermore, because the bialkali material has high SEY, free electrons emitted from the cathode edge can trigger the MP, so the size of the photocathode layer must be limited and the rest of the surface must be shielded during evaporation. The typical QE after a 2.5 h preparation process is around 10%, which can be reserved well under an ultrahigh vacuum of 8 × 10^–11^ torr in the transport suitcase. The cathode QE experiences slight degradation during the transfer and insertion and finally stabilizes at about 5% when the photoemission begins in the SRF gun. Figure 7 shows the cathode QE evolution in the gun operating in CW RF mode with a pulsed beam, where one can see the strong influence of the MP on the cathode performance during the gun operation.

#### 5.3.2. K_2_CsSb on Mo Plug in PKU SRF Gun

Another successful application of the reflection-mode K_2_CsSb photocathode is in the PKU DC-SRF photoinjector II, which generated a pulsed beam with a bunch charge of ∼150 pC and a repetition rate of 1 MHz in the first commissioning [44].

The special design of the DC structure plus the 1.5-cell niobium cavity can eliminate the compatibility problem between the SC cavity and the NC cathode (Figure 8). Because the cathode temperature is kept at 36 K during operation, this experiment is the first one to operate the K-Cs-Sb photocathode below the temperature of liquid nitrogen. However, a strong degradation of the QE was observed at 36 K during commissioning.

D. Ouyang and collaborators developed a new recipe to shorten the duration of the cesiation process, which proved to be helpful for increasing the lifetime of K-Cs-Sb [45]. However, the K_2_CsSb photocathode in the gun provided only a low QE of 0.3%, which was less than that measured at room temperature in the suitcase. Thus, the operation temperature of the photocathode should be kept higher than in this experiment if higher QE is needed for beam operation.

#### 5.3.3. K_2_CsSb on Mo Plug for HZB SRF Gun

Reflection-mode K_2_CsSb has been also planned for the 1.5-cell SRF gun developed at Helmholtz Zentrum Berlin. The cathode is placed inside the RF filter, which is cooled with 80 K gaseous helium (Figure 9). Because this gun is designed for high-current purposes, the most important key for successful operation is the sufficient cooling of the cathode, where a heat load on the photocathode as high as 30 W is expected [46]. Otherwise, cathode overheating could lead to sequential consequences, starting with photocathode evaporation, cathode plug destruction, and even causing the quality degradation of the superconducting cavity.

In the HZB photocathode lab, the co-deposition of K and Cs on Sb has been successfully demonstrated. This method is believed to precede the sequential growth procedure [47]. In situ X-ray photoelectron spectroscopy (XPS) was applied to optimize the growth process of K_2_CsSb on Mo. It is worth noticing that the cathode performance must not be influenced by cryogenic operation temperatures. In the HZB study, two K_2_CsSb cathodes were cooled down to 150 K and 120 K, respectively, and their QEs were not reduced when excited with green photons (2.4 eV) (Figure 10).

### 5.4. Transmission-Mode K_2_CsSb for KEK SRF Gun

The distinguishing feature of the KEK SRF gun is its cathode substrate made of the transparent substrate MgAl_2_O_4_ and transparent superconductor LiTi_2_O_4_, which enable a thin K_2_CsSb film able to work in the transmission mode (Figure 11). The transmittance of the substrate is about 70% at a wavelength of 477 nm. LiTi_2_O_4_ is an epitaxial thin film deposited on MgAl_2_O_4_ (111) because the RF penetration depth of superconductor LiTi_2_O_4_ is about several tens of nanometers. The idea is to use the transparent superconductor LiTi_2_O_4_ to block the RF leakage and transmit the cathode-driven laser onto the backside of K_2_CsSb. Furthermore, the lattice constant of LiTi_2_O_4_ (0.8405 nm) matches that of K_2_CsSb crystal (0.861 nm). The transition temperature of LiTi_2_O_4_ is 13 K, so the cathode and its holder will be cooled with liquid helium down to around 2 K [49].

T. Konomi and his team successfully prepared K_2_CsSb on SiTiO_3_ during commissioning and achieved 10% quantum efficiency at 405 nm at room temperature [49]. However, the QE decreased when the cathode was cooled down to liquid helium temperature, but this QE–temperature correlation was not reversible. The residual gas, absorbed onto the cathode surface, might result in the contamination of the cathode surface. With this cathode, the initial emittance at cryogenic temperature was measured with 405 nm, about 0.6–0.65 µrad/mm. Further study on the K_2_CsSb on SiTiO_3_ or LiTi_2_O_4_ at cryogenic temperatures is essential for the final successful application.

### 5.5. III-V Photocathodes: GaAs, GaN

Polarized electron beams are required in many high-energy nuclear physics experiments. Currently, the only photocathode that can provide such a polarized electron beam is GaAs. However, its negative electron affinity (NEA) surface is extremely sensitive to the residual gas. Up to now, a GaAs photocathode has been applied only in DC guns. Because SRF guns are able to provide a super clean environment and ultra-high vacuum due to the cryo-pumping effect on the cavity wall, it is possible to apply GaAs photocathodes in SRF guns.

Ten years ago, BNL planned to apply GaAs(Cs,O) in a 1.3 GHz half-cell SRF gun. A commercial 100 µm GaAs photocathode was installed using indium into a recessed Nb plug and successfully activated with a yo-yo process or Cs-O simultaneous activation, and a QE of 3–10% at 532 nm was obtained in the cathode preparation chamber [50,51] but dropped down to 0.8% after passing through a transfer section. The SC cavity was tested with GaAs inside and the heat load generated on GaAs was studied as well [52]. However, the first test showed unsatisfactory results, and no further progress on this project has been published. Recently a new plan for a polarized SRF gun was published by BNL [42]. A GaAs photocathode was proposed to be applied in the 113 MHz QWR SRF gun, based on the rich experience of K_2_CsSb operation in the 113 MHz gun. It will be a promising program.

Different from the GaAs photocathode, GaN is not applied to provide a polarized beam. It becomes attractive for high brightness photoinjectors because of its high QE at the third harmonic wavelength (for example 340 nm) and it is as robust as Cs_2_Te. There are already discussions to use GaN in a DC gun [53]; actually, it is also a new idea for SRF gun application.

J. Schaber et al. have started to characterize this material as a potential candidate for HZDR SRF guns [54]. In the first experiments with the commercial p-type GaN samples on sapphire or silicon, the activation process was successful and the QE could be reserved after cathode transfer. The cleaning procedure should generate an atomic clean surface for efficient activation, so it has a big influence on the final QE [55]. The proper thermal cleaning process can greatly improve the QE and lifetime of the same GaN cathode.

In order to understand and follow the degradation of the cathode QE and the evolution of the NEA surface, HZDR is using an XPS to study the surface chemical composition. However, there is still a long way to go before GaN could be successfully used in an SRF gun; for instance, a conductive substrate is requested for high current extraction. At the moment, the sapphire substrate as an insulator is not the best choice to ensure the high electrical and also thermal conductivity of the GaN photocathodes.

## 6. Conclusions and Discussions

The successful operation of all photoinjector types relies on good cathode quality and suitable cathode insertion. Choosing the right cathode and integrating it into the superconducting cavity are the essential steps during the design of a superconducting RF photoinjector. The answers depend strongly on beam requirements, gun cavity type, and developing phase.

In recent years, photocathode work for SRF guns has achieved remarkable progress. In this review, we focused primarily on the cathode application in an SRF gun environment. Although material science and photoemission theory are not the main concerns here, they are also very important topics for cathode development. We show various successful solutions for cathode integration as well as those photocathodes used or proposed for SRF guns. Table 2 summarizes the operation properties of photocathodes for present SRF guns. The SC photocathodes Nb and Pb were proposed for several gun projects, but no operation report has been published up to now. If a new gun uses an NC photocathode, a metallic cathode such as copper is a good choice for commissioning purposes. Magnesium cathodes can be selected for moderate current applications. Meanwhile, semiconductor photocathodes have been proved safe for SRF guns. Both Cs_2_Te driven by a UV laser and K_2_CsSb driven by a green laser can be reliably operated with satisfactory performance in SRF guns at HZDR and BNL. However, the experience of the PKU SRF gun and KEK SRF gun showed that the QE of the K_2_CsSb cathode at cryogenic temperature decreased dramatically, so more studies are needed. Furthermore, new plans to apply commercially available III-V semiconductors, such as GaAs and GaN, have been reported, and some preliminary studies have already been published.

The operation temperature of a photocathode in an SRF gun is worth being noticed. They vary from 2 K to 300 K (shown in Table 2). Theoretically, the temperature has an impact on the lattice size and the scattering rates of electrons and thus on the intrinsic emittance of the photocathode [56]. Practically, the main aspect of various operation temperatures is the difference in vacuum environment due to the cathode temperature. In the case of a warm photocathode in an SRF gun, the cavity works as a high-speed cryogenic pump, so the photocathode faces an optimal vacuum to main the surface cleanness. Oppositely, if the cathode works in a cryogenic status, there might be a risk of the cold photocathode adsorbing rest gases. From this point of view, keeping the cathode warmer than the cavity can benefit the cathode to maintain its QE and a long lifetime.

As is well known, not only systemic scientific study is important for the development of photocathodes, but also technical knowledge and experience are sometimes the decisive factors for success. Especially, expertise is needed to solve some critical issues, such as QE reserving, vacuum during transport and operation, heat load, sufficient cooling, suppressing MP, and so on. We hope this review can provide some useful references from cathode applications with respect to the developers or users of SRF photoinjectors.

## Figures and Tables

**Figure 1 micromachines-13-01241-f001:**
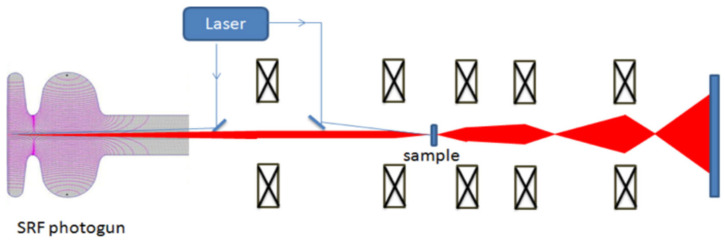
Schematic layout of the Euclid SRF photogun-based UEM system [14].

**Figure 2 micromachines-13-01241-f002:**
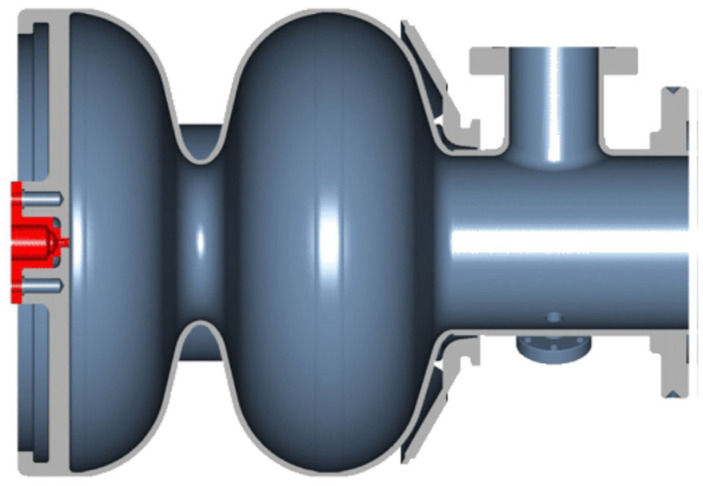
DESY SRF Gun cavity and its superconducting cathode plug [12].

**Figure 3 micromachines-13-01241-f003:**
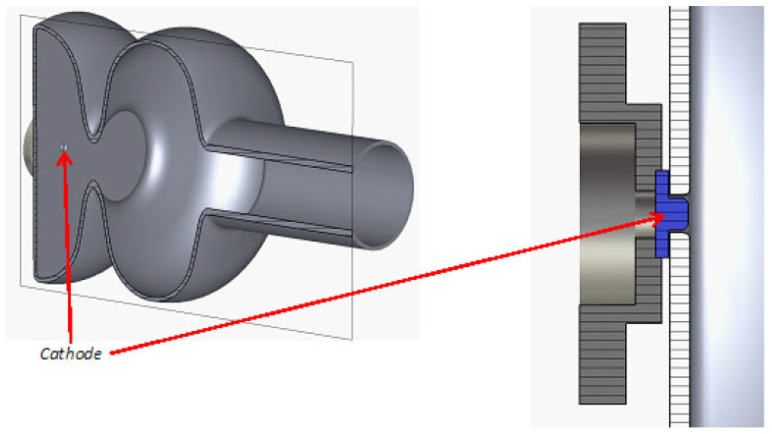
Design of Jlab/RadiaBeam SRF gun with plasmonic Nb cathode [13].

**Figure 4 micromachines-13-01241-f004:**
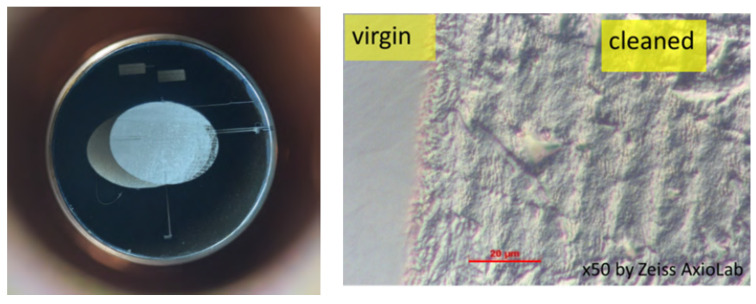
Photograph of Mg photocathode Mg#214 after second laser cleaning (**left**) and surface structure changes after cleaning (optical microscope image, (**right**)) [29].

**Figure 5 micromachines-13-01241-f005:**
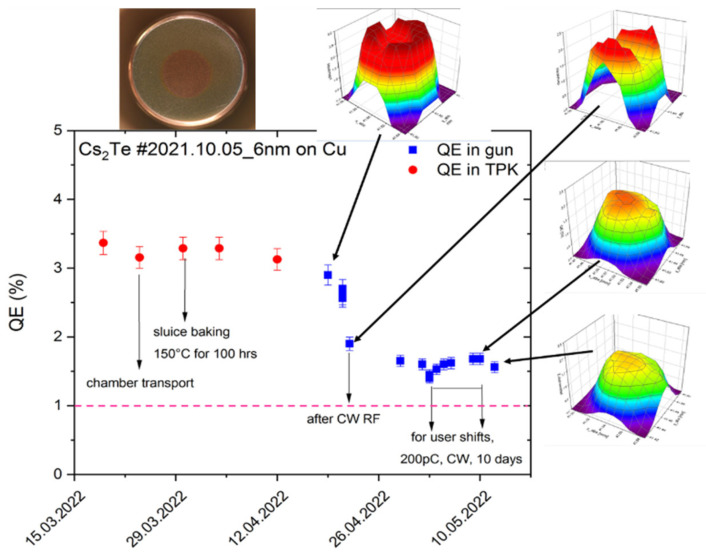
Performance of Cs_2_Te photocathode during cathode storage in SRF gun. Insertion: Cs_2_Te film on a copper plug and the QE mapping on different days [39].

**Figure 6 micromachines-13-01241-f006:**
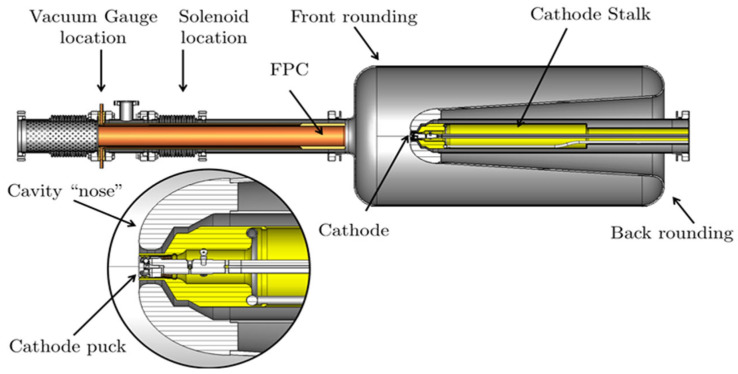
Cross-section of the BNL QWR SRF gun with cathode stalk [17].

**Figure 7 micromachines-13-01241-f007:**
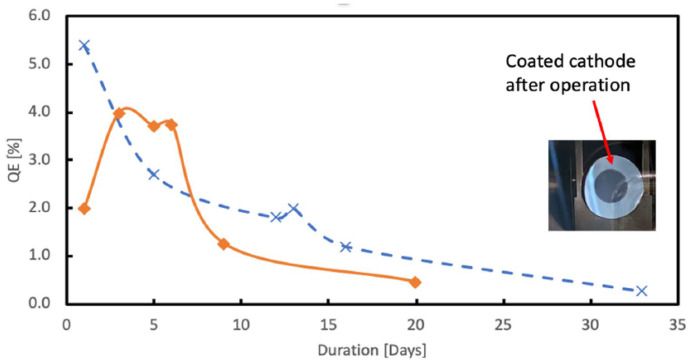
The cathode QE evolution in the BNL SRF gun with CW RF but a pulsed beam. Solid orange curve: the cathode exposed to MP; dashed blue curve: the cathode not exposed to MP [42].

**Figure 8 micromachines-13-01241-f008:**
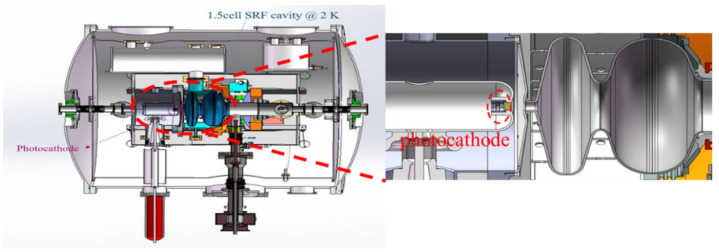
Design of DC-SRF photoinjector II. The photocathode is installed in the DC structure [44].

**Figure 9 micromachines-13-01241-f009:**
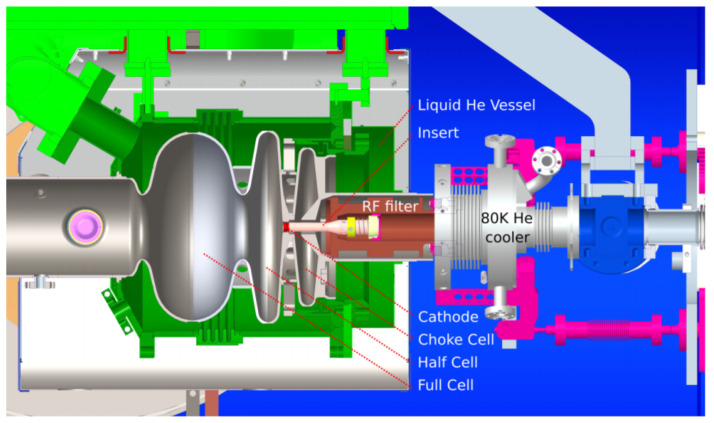
Cross-section of the HZB SRF photoinjector. The cathode is installed in the RF filter and cooled with liquid nitrogen [48].

**Figure 10 micromachines-13-01241-f010:**
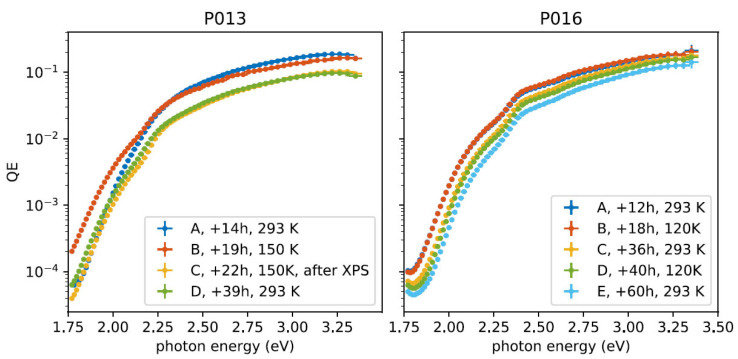
Spectral response of K_2_CsSb cathodes (sample #P013 and#P016) during the cryogenic cooling experiments performed in the HZB photocathode preparation chamber [47].

**Figure 11 micromachines-13-01241-f011:**
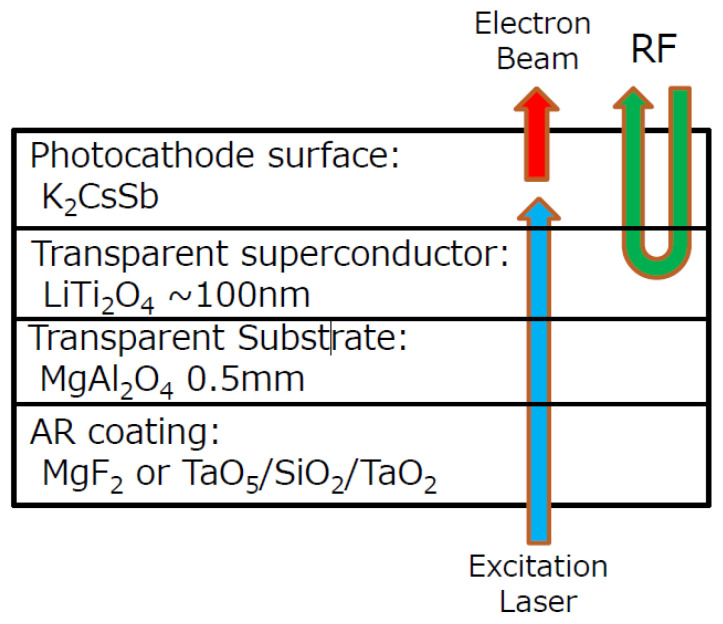
Working concept of transparent photocathode for KEK SRF gun [49].

**Table 1 micromachines-13-01241-t001:** Several stable metals with a work function of less than 5 eV considered as photocathodes in electron guns.

Metal	QE	ϕ (eV) [28]
Cu	10^−5^–10^−4^	4.6
Mg	10^−5^–10^−3^	3.6
Nb	10^−6^–10^−4^	4.3
Pb	10^−6^–10^−3^	4.0

**Table 2 micromachines-13-01241-t002:** Operation properties of photocathodes for SRF guns.

Cathode Material	Cathode Type	Drive Laser (nm)	Quantum Efficiency in Operation	Lifetime	Extracted Bunch Charge	Operation Temperature	Ref.
Nb	SC metal	266	10^–5^ *	No limit *	<1 pC **	4 K	Euclid [14] JLab [19]
Pb	SC metal	213	10^–4^–10^–3^ *	Year *	20–250 pC **	2 K	DESY [12,22]
Pb	SC metal	258	9 × 10^–5^		100 pC **	1.8 K	HZB [21]
Plasmonic Nb	SC	1030	1.4 × 10^–5^ *			2 K	Jlab [13]
Cu	NC metal	258	10^–5^	No limit	~1 pC	80 K	HZDR [26] HZB [16]
Mg	NC metal	262	10^–3^	year	250 pC	80 K	HZDR [26]
Cs_2_Te	Semiconductor	262	1~3%	~3 months	300 pC	80 K	HZDR [39]
K_2_CsSb	Semiconductor	532	5–10%	>1 month	19.7 nC	300 K	BNL [42,43]
K_2_CsSb	Semiconductor	519	0.3%		100 pC	36 K	PKU [44,45]
K_2_CsSb	Semiconductor	515	0.7–5% *	5 days	77 pC **	80 K	HZB [46,47,48]
K_2_CsSb Transmission-mode	Semiconductor	532	~2% *		80 pC **	2 K	KEK [5,49]
GaAs	III-V semiconductor	532	0.7% *			2 K	BNL [50,51,52]
GaN	III-V semiconductor	310 **	>1% *			80 K	HZDR [54,55]

* Expected value. ** Design Value.

## Data Availability

Not applicable.

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
