# Peer review of "Review of Recent Progress on Advanced Photocathodes for Superconducting RF Guns"

_micromachines, 2022, doi:10.3390/mi13081241_

Round 1

Reviewer 1 Report

This review paper has covered various photocathodes used in SRF guns. The authors provided a deep insight into photocathode development and testing results. The discussion of the possible improvement of photocathode for future SRF guns is very valuable. Meanwhile, I think more detailed discussions will help to improve the quality of this paper. I suggest publishing this paper after addressing my comments.

One comment about the title: Better spell out SC RF. Not all readers know these acronyms. Same concerns on many acronyms in the paper. These acronyms must be spelled out when shown the first time: e.g. FEL, ERL, DC, QWR,UEM/UED,QE,UV,MP. Forgive me if I missed other acronyms used in the paper.

  1. line 94-95: The authors discussed integrating NC cathode by using a DC gun out of SC cavity. It sounds to me this kind of setup should belong to DC gun, other than SRF gun. Because the cathode is in the DC field, other than in RF field. Maybe in the introduction section, give a clear definition: What is SC RF gun. Then based on the definition, the authors can decide DC gun plus SRF cavity shall be fitted into this paper.

  2. line 123: Comment on entire subsection 4.2: With the laser illuminates on the Nb or Pb superconducting cathode. Laser power may heat the cathode. Could you comment that the Nb/ Pb is still superconducting? If not, should still call it SC cathode? I think there must be a limit on laser power density. Once above such a power limit, the gun will either quench or have a significant Q0 drop. For this concern, can you comment on SC cathode application?

  3. line 147: Comment on subsection 4.3: The plasmonic cathode has a nonlinear QE, which needs multiple photons involved. Typically, the higher the laser pulse energy, the higher QE. If the laser pulse energy is low, the nonlinear QE may be even lower than the linear QE. However, higher laser pulse energy will cause high average power if SRF gun operates in high rep. rate. This kindly contradicts with SRF cathode requirement if using the plug cool method. Could you add clear motivation for this research or state why you think this proposed method is promising?

  4. line 190: I suggest improving Figure 4's resolution. It is difficult to see all the elements. Also, this figure shows the calculated work function is well matched to the measured value. Is this important(relevant) to this paper? What you want to deliver are various metal materials that have different work functions. You would like to pick the metals with lower work function, high chemical stability, and are easy to integrate into SRF gun. Maybe a better figure with only measured work function (or calculated work function) versus material or a table will be clearer.

  5. line 247-253: The QE drops from 3 % to about 2 % (almost 30% of initial QE) is when CW RF loaded. If I understand, at this time, the laser is still off. (If I misunderstood, then please clarify in the paper). But in line 254, the author explains the QE degradation due to dark current or the beam halo. Usually beam halo is either from laser halo or betatron mismatch due to high space charge. If no laser is involved, where does beam halo come from? Yes, I agree dark current is one reason. Is it possible the RF starting up process MP (line 261) caused the wall to release the absorbed gases? I like your explanation of ionizing the released gas molecules.

  6. line 403: figure 13. In the figure, you have A4, A5, A6. You may have to show the meaning of the label in this figure. If they are not important, then please consider removing them.

Reviewer 2 Report

The paper provides a review of the recent progress in the development of photocathodes for superconducting RF (SRF) photoinjectors. The concept of an SRF gun has always been considered promising in theory but challenging in execution. Introduction of a photocathode into the body of an SRF gun creates multiple complications not only in terms of the preservation of the cathodes’ QE and lifetime, but also the operation and performance of the gun itself. Development of robust high-QE photocathodes is critical for the advancement of SRF gun technology. The manuscript provides an overview of the available photocathode materials and techniques used to produce the cathodes. The authors include an overview of the different cathode choices, methods of cathode introduction into the cavity body, and experimental experiences worldwide. The paper is well-structured; however, the manuscript requires extensive editing of the language and style. There are numerous typos that need to be addressed. Below I outline some questions and suggestions for the authors along with some (but not all) typos that I have caught while reading through the paper. I find the presented review valuable to the accelerator community (specifically groups focused on the SRF gun and cathode development). I recommend the manuscript for publication after the corrections are made.

Questions and suggestions:

1.     Line 24: provide some references for the DC and NC guns.

2.     Line 95: provide a reference for the PKU DC-SRF gun.

3.   Please, consider providing a paragraph summarizing advantages and disadvantages of cold vs warm photocathodes. Some statements are scattered throughout the manuscript, but it would be beneficial to clearly outline the benefits and pitfalls of both.

4.     Figure 1: what do the letter notations at the bottom of the figure stand for? If it’s not critical for the paper, they can be simply removed.

5.     Figure 6: Axes labels of the QE map insets are unreadable. Please, consider enlarging the images and increasing the font size.

6.     Line 288: BNL SRF gun not only has the RF fingers, but also a half-wave RF choke with an impedance transformer.

7.     Line 290: “E. Wang described the detailed preparation process”. I’m guessing, you mean the process for the cathode growth.

8.     Please, consider adding a table that will summarize the major parameters of the cathodes presented in the manuscript. It will allow for an easy comparison between the different methods used (list the cathode material, laser used, warm/cold, etc.) and parameters demonstrated experimentally (QE, lifetime, max extracted bunch charge, etc.).

9.     Figure 11 is missing y-axis label and numbers.

10.  Line 196: Please, try to be more specific: what does it mean to have “incredibly long lifetime” and “only little MP problem”? If possible, provide some ballpark for the lifetime and elaborate on the MP.

Typos and minor details:

1.     Line 10: typo – “radio frequency”, not “ratio”

2.     Line 15: “improvement from cathodes side” needs to be rephrased. 

3.     Line 21: typo – “radio frequency”, not “ratio”

4.     Line 30: change “to use a superconducting material…, into the cavity back wall ” to, for example, “to insert a superconducting material…, into the cavity back wall ”

5.     Line 36: “enough QE” please rephrase, for example “high enough QE”

6.     Line 47: “in application to the SRF guns”

7.     Line 66: the word “deposed” is not applicable here.

8.     Line 67: “if no sufficient cooling is for the cathode” needs to be rephrased.

9.     Line 71: “resonated electrons” needs to be rephrased.

10.  Line 89: electrically isolated.

11.  Line 94: outside of the SC cavity.

12.  Line 97: I believe, Pierce is a proper noun and needs to be capitalized. Please, provide some references for the Pierce-type electron sources.

13.  Line 116: RRR is not defined in the manuscript.

14.  Line 124: “The All Superconducting RF gun” – needs to be corrected.

15.  Line 137: remove the word “were”

16.  Line 149: superconducting photocathode material

17.  Line 154: “There are different approaches”

18.  Line 165: “At the moment”

19.  Line 172: “At the moment”

20.  Line 223: remove the word “to”

21.  Line 288: “gold plated RF spring”

22.  Line 295: “the rest of the surface”

23.  Line 298: “slight degradation”

24.  Line 389: “are as robust as Cs2Te”

25. Please, pay attention to the formatting of the references. Some titles are written in all caps, some are not.
